# UV-degraded polyethylene exhibits variable charge and enhanced cation adsorption

Ryan Bartnick[1]*, Shahin Shahriari[1], Günter K. Auernhammer[2], Ulrich Mansfeld[3], Werner Reichstein[4], Lisa Hülsmann[5], Eva Lehndorff[1]

1 Soil Ecology, Bayreuth Center of Ecology and Environmental Research (BayCEER), University of Bayreuth, Bayreuth, Bavaria, Germany, 2 Polymer Interfaces, Leibniz-Institut für Polymerforschung Dresden e.V., Dresden, Germany, 3 Keylab Electron and Optical Microscopy, Bavarian Polymer Institute, University of Bayreuth, Bayreuth, Bavaria, Germany, 4 Ceramic Materials Engineering, University of Bayreuth, Bayreuth, Bavaria, Germany, 5 Ecosystem Analysis and Simulation, Bayreuth Center of Ecology and Environmental Research (BayCEER), University of Bayreuth, Bayreuth, Bavaria, Germany

* ryan.bartnick@uni-bayreuth.de

## Abstract

The widespread use of plastics has led to an omnipresence in soils. We aim to understand whether transformation of polyethylene (PE) and polyethylene tere-phthalate (PET) in the atmosphere alters their surface properties which, after input of microplastics to soil, leads to an increase of reactive surfaces in soils. PE and PET particles (sieved 200–400 µm) were exposed to accelerated UV degradation. Changes in particle size and surface morphology were measured (using electron microscopy) and compared to pH-dependent variation in surface charge parameters (zeta potential and cation exchange capacity). Fourier transform infrared spectroscopy and X-ray photoemission spectroscopy detected the formation of functional groups and surface atomic composition. After 2000 hours of degradation, PE particles reduced in size from $375 \pm 117$ µm to $8 \pm 7$ µm, while PET particles showed only a slight decrease in size, from $653 \pm 219$ µm to $484 \pm 274$ µm. Reduction of particle sizes correlated with increased absolute zeta potential and a decrease of the isoelectric point. Hydrated surface charge of degraded PE after 2000 hours was unstable under alkaline conditions, related to the formation of carbonyl groups on its surface and increase in hydrophilicity. PET showed fewer surface chemical changes. Especially for degraded PE incorporated in soil, the alteration of its surface can exhibit comparatively one-tenth of the cation sorption power of clay in alkaline environments ($\approx 7.5$ vs. 77 $cmol_c$/ kg at pH 9), while degraded PET remained comparatively low ($\approx 1.1$ $cmol_c$/ kg). This study demonstrates that PE undergoes substantial physico-chemical changes during UV degradation, increasing its reactivity, while PET remains relatively stable. These findings highlight the need for further studies to differentiate and understand the effects of diverse plastic types on soil ecosystems.

**Data availability statement:** All data files are available from the Zenodo database (https://doi.org/10.5281/zenodo.15571270).

**Funding:** Funded by the Open Access Publishing Fund of the University of Bayreuth. Funded by the Deutsche Forschungsgemeinschaft (DFG, German Research Foundation) – SFB 1357 – Project Number 391977956. The funders had no role in study design, data collection and analysis, decision to publish, or preparation of the manuscript.

**Competing interests:** The authors have declared that no competing interests exist.

## Introduction

Poor waste management and high plastic production have led to a significant accumulation of plastic waste in soils where nutrient and carbon storage can be affected [1]. The impact of plastics as foreign reactive surfaces on soil functions remains poorly understood, as the change of plastic surface properties during environmental exposure must be considered. The environmental fate of microplastics (MPs) depends on their chemical composition, degradation state, and interaction with other environmental particles. Among all sinks for MPs, agricultural soil may be a main hotspot for MPs pollution because of intensive agricultural activities [2].

Polyethylene (PE) is the most widely used plastic in the world, accounting for around 30% of all plastics [3]. PE degrades relatively easily when exposed to ultraviolet (UV) light [4]. Polyethylene terephthalate (PET) is also a widespread plastic and of environmental concern due to its strong resistance to degradation [5]. Globally, 79% of produced plastics are estimated to remain on land, accumulating in landfills or the natural environment [3]. When exposed to UV light, PET can break down into smaller particles, but this process is slow, and the particles can persist in the environment. For these reasons, PE and PET were chosen in this study to understand the physicochemical transformations during UV degradation.

One key problem with plastic pollution is the breakdown of larger plastics into MPs and nanoplastics (NPs). As plastics degrade, they break down into smaller pieces due to sunlight, temperature changes, physical abrasion, and biological processes [6], causing their properties to change and affecting their behavior in the environment [7,8]. Size reduction increases the number of particles and their surface area, enhancing their environmental mobility and interaction potential, however, this still needs to be related to surface properties of MPs.

To understand how degraded MPs interact with complex environments like soil, it is essential to determine how degradation processes alter the physical and chemical properties of pristine (non-degraded) MPs. Photodegradation, driven by exposure to UV radiation, leads to the breakdown of polymer chains and the formation of new functional groups such as carbonyls and hydroxyls on the particle surfaces, which increase their reactive surface [7,8]. Fourier-transform infrared spectroscopy (FTIR), a powerful analytical technique widely used to identify functional groups and chemical bonds in organic compounds, shows these chemical changes, indicating oxidative degradation [9]. By analyzing the specific wavelengths of light absorbed by a sample, FTIR can provide detailed information about molecular vibrations, providing analysis to changes to plastic molecular composition throughout UV degradation. Additionally, X-Ray photoelectron spectroscopy (XPS) can be used to provide surface chemical composition of polymeric structures at 2–10 nm sampling depth at high precision [10]. By observing the bonding energies of emitted electrons from X-ray irradiation of the polymer surface, molecular information can be obtained to characterize changes in degraded polymers. Changes in plastic surface chemical speciation can be detected throughout the plastic degradation process.

To evaluate the change in surface charge of degraded MPs, the zeta potential of degraded PE and PET MPs can be analyzed to understand surface charge

variations. Zeta potential, or electrokinetic potential (symbolized as ζ), is the electrical potential difference across the mobile portion of the electrical double layer surrounding a colloidal particle. The isoelectric point (IEP), or point of zero charge, is a crucial marker for understanding how particles behave in solutions of a specific pH, because it gives the pH at which the surface appears neutral to the surrounding. Instead of using zeta potential, the IEP allows seeing if particles have potential to react with other soil constituents [11].

While pristine plastics may be considered inert and exhibit no ion exchange, degraded plastics have more potential to significantly influence soil properties, e.g., cation exchange capacity (CEC), which is vital for nutrient retention and availability in soils. The CEC of soil measures the quantity (moles) of negatively charged sites on soil particles that attract positively charged ions, such as soil nutrients. Degraded MPs could incorporate and aggregate with soil components, potentially changing soil structure and chemical properties [12]. Alterations in CEC can be attributed to complex and various factors, including the physicochemical traits of MPs, their interaction with soil particles and microorganisms, and changes in soil pH and organic matter content caused by MPs [13,14]. Soil pH has a significant impact on the effective CEC, changing the quantity and direction of charge of soil constituents (mainly OM and oxidic phases). Whether degraded MPs of decreasing size may alter soil CEC by varying the amount of charged surfaces due to their increased surface area needs to be studied.

The main goal of this study was to understand how the UV degradation of PE and PET influences their surface chemistry and adsorption capacity for ions that potentially affect the quality of soil. We investigated if a UV degradation process significantly altered the size, surface morphology, surface charge, and chemical composition of PE and PET using a combination of light microscopy, scanning electron microscopy (SEM), as well as their potential to affect cation exchange as expected in soils. In this study, we quantified the change in size of PE and PET during degradation and visualized changes in hydrophobicity and roughness with ESEM; we used FTIR and XPS to identify functional group formation on the surface of degraded PE and PET; and we quantified the alteration of CEC and zeta potential of degraded PE and PET. The polymers used in this study had an initial sieved size of 200–400 μm and were degraded for up to 2000 hours by UV radiation under controlled, moist conditions.

## Materials and methods

### Experimental plastics, materials, and artificial degradation

The plastic materials used in this study were low density-polyethylene (LD-PE, Lupolen 1800 P-1 – LyondellBasell, Rotterdam, NL) and polyethylene terephthalate (PET, CleanPET WF – Veolia Umweltservice, Hamburg, Germany). Further technical data sheets of plastic polymers are available online (see Data Availability). Plastics were initially prepared by cryomilling (ZM200; Retsch, Haan, Germany) and airjet sieving (E200 LS; Hosokawa Alpine, Augsburg, Germany) to a size range of 200–400 μm of irregular shape. All materials were kept in glass containers and handled with metal utensils.

In a Q-SUN XE-3 accelerated degradation chamber (Q-LAB, Westlake, OH) equipped with three xenon lamps and a Daylight-Q filter, MPs samples were subjected to various conditions to simulate natural degradation processes [8,15]. MPs were exposed to UV radiation to mimic solar radiation (UVA), irradiated with 60 W/m$^2$ (at 300–400 nm), corresponding to a total irradiance of 594 W/m$^2$, comparable to the spectrum of natural sunlight, with an estimated accelerated degradation 5x faster than in the environment [16]. MPs were maintained at a constant temperature of 38°C, immersed in deionized water, and under mechanical stress from stirring with a PTFE-coated magnetic stir bar. The continuous stirring ensured that the particles were uniformly irradiated from all sides. To study how degradation affects surface changes and ion adsorption, three sets of both PE and PET samples were produced: non-degraded (0 hours), and exposed samples in an accelerated degradation chamber for 400 and 2000 hours. According to Menzel et al. [16], this corresponds to approximately 3 and 14 months of environmental degradation, and these exposure times were selected based on previous studies that observed notable physicochemical changes in MPs at these durations.

## Size and visual characterization of MPs

Scanning electron microscopy was performed at 3 kV on a LEO 1530 (Zeiss, Oberkochen, Germany). Prior to imaging, the particles were coated with a thin layer of platinum using a Cressington 208 HR sputter coater. ESEM was performed on a FEI Quanta FEG 250 (Thermo Fisher Scientific, Waltham, USA) equipped with a cooling stage and a gaseous secondary electron detector (GSED). The pristine samples were placed on a polished graphite support and cooled down to 2 °C at 400–600 Pa for 30 min. For the wetting experiments, the pressure was increased to a range of 725–800 Pa (rate 600 Pa/min) depending on the wettability, and wetting was imaged at 10 kV at constant pressure.

The size and distribution of particles were measured using images taken by SEM and a transmitted light ECHO Revolve microscope with 10x magnification, which allowed detection of particles with a lower limit around 1 μm in size. In each image ($n_i = 4$–37), the diameter of each particle was measured ($n_p = 107$–861) using Fiji 2.9.0 image processing software [17], and then frequency distribution graphs were produced using OriginPro 2024b software (OriginLab). Due to the random, oblong shapes of particles following size fractionation, diameter of each detected particle was measured at the longest and shortest sides that could pass through the airjet sieve, which results in a larger detected distribution then what is expected from airjet sieving which can pass particles that are narrower than 400 μm on one side but longer on another.

## Surface chemical characterization of MPs

FTIR detected functional group change in the plastic surfaces at a penetration depth of few micrometers, while XPS measured at higher resolution at the top surface of a few nanometers. The FTIR surface analysis of PE and PET MPs was conducted using an Alpha II spectrometer (Bruker, Billerica, USA) equipped with a diamond ATR crystal. The MPs samples were prepared by placing dry particles directly onto the analyzer without any specific sample preparation. The FTIR measurements covered a range of wavenumbers from approx. 4000–400 cm$^{-1}$ with a resolution of 4 cm$^{-1}$, averaging 8 sample scans and 8 backgrounds scans. This allows quick identification of changes in the IR spectra of MPs after degradation.

XPS measurements were taken with a PHI 5000 VersaProbe III (ULVAC-PHI, Chigasaki, Japan), peaks identified by binding energy [18], and spectra produced with MultiPak 9.8.0.19 software. Plastic samples were fixed on a sample holder with double-sided tape. Measurements were taken in triplicate as a collection, which were neutralized (electron and argon) and binding energy corrected. Excitation energy was monochromatic aluminum K-alpha. Surveys for elemental composition were taken as a scan (pass energy = 224 eV, step size = 0.8 eV), and curve fitting of C1s peak was performed with a high resolution, detailed spectrum (pass energy = 26 eV, step size = 0.1 eV). XPS survey spectra were inspected for adventitious elements, and a low-intensity F1s signal was observed on the subset of 2000 h samples (see Table 1). Given the use of PTFE-coated stir bars during long exposures, PTFE abrasion from stirring is the most probable source of surface fluorine contamination.

**Table 1. XPS element ratios of the surface composition of PE from 0 to 2000 hours degradation.**

|  | element ratios |  |  |
| --- | --- | --- | --- |
| plastic | O/C | Si/C | F/C |
| PE 0 h | 0.023 | 0.012 | ND |
| PE 400 h | 0.069 | 0.029 | ND |
| PE 2000 h | 0.110 | 0.040 | 0.026 |
| PET 0 h | 0.348 | 0.011 | ND |
| PET 400 h | 0.291 | 0.012 | ND |
| PET 2000 h | 0.340 | ND | 0.105 |

ND = Not Detected

## Streaming zeta potential

The surface potential (ψ) of particles can be calculated by using the experimentally measured zeta potential (ζ), although the actual zeta potential typically remains lower than the surface potential calculated from the diffuse double-layer theory. Zeta potential shows the difference in potential between the shear plane and the bulk solution [19,20]. In the streaming zeta potential process, a solution is pushed through a capillary channel with a specific applied pressure. The SurPASS 3 (Anton Paar, Graz, Austria) was used to measure both streaming potential and streaming current [21].

For these measurements the respective amount of powder was fixed with 20 μm membranes, pore size chosen by the size distribution values, in the powder sample holder. This part was inserted in the cylindrical cell of the instrument equipped with Ag/AgCl-electrodes. The permeability index was adjusted around 100. The measuring fluid was streamed through this powder plug in the pressure range from 600 to 200 mbar. The zeta potential ζ was calculated according to the Smoluchowski equation:

$$\zeta = \frac{dU}{dp} \times \frac{\eta \kappa}{\varepsilon_r \, \varepsilon_0}$$

(1)

Where $U$ is the streaming potential, $p$ is the pressure loss, $\varepsilon_r$ and $\varepsilon_0$ are the dielectric constant and the vacuum permittivity, $\eta$ is the viscosity and $\kappa$ the conductivity of the measuring fluid. The pH-dependence of zeta potential or the powder was determined in the presence of KCl solution, concentration of $10^{-3}$ mol/L, as function of the pH value. We started at pH~6 and adjusted the pH value by stepwise adding HCl or KOH. By looking at the shape of the zeta potential versus pH curves and where the zeta potential is zero (at the IEP), we can understand the types of the functional groups on the surface of the fibers (see S1 Fig); for non-polar or non-dissociating surfaces, the IEP was determined around pH 4. To compare the absolute values accurately, we needed to ensure the same testing conditions. We made efforts to maintain consistent flow conditions, but because of differences in particle structure after degradation, the weight and surface area varied between PE and PET. PE, with its more fibrous structure, could be compressed strongly with a smaller sample amount, while the PET's particle-shaped structure required a larger sample amount to achieve similar flow properties. We note that the IEP is only little influenced by these changes in permeability.

## Cation exchange capacity determination

To determine CEC, we followed modified methods by Liu et al. [22] and Schäfer & Steiger [23]. In addition to testing degraded and non-degraded MPs, we also included adsorption to montmorillonite clay as a reference material for highly charged soil particles and a control of pure, inert quartz sand. Potential CEC is measured as strontium ($Sr^{+2}$) through a reverse desorption reaction via replacement of the sites via magnesium ($Mg^{+2}$). Initially, cation and material equilibration occurred by percolating the sample with 0.1 M strontium chloride-triethanolamine buffered at pH 4, 7, and 9 (pH adjusted with HCl or KOH, column filled with 0.1 to 0.4 mm quartz sand, pre-cleaned by rinsing with acetone:cyclohexane (1:1) and heating to 900 °C). Triethanolamine helped disperse MPs and minerals.

The magnesium chloride ($MgCl_2$, 0.1 M) solution from the reverse exchange was filtered through a 0.45 μm cellulose acetate filter and analyzed for total Sr using inductively coupled plasma-optical emission spectroscopy (ICP-OES 5800, Agilent Technologies, Waldbronn, Germany). The number of charges from $Sr^{2+}$ ions in the volumetric flask was estimated as the CEC of the sample. This value would be equivalent to the CEC $cmol_c$/ kg of our sample as follows:

$$\frac{mg\ Sr^{2+}}{kg\ sample} \times \frac{1\ g}{1000\ mg} \times \frac{1\ mol\ Sr^{2+}}{87.62\ g} \times \frac{100\ cmol}{1\ mol} \times \frac{2\ cmol_c\ valence\ Sr^{2+}}{1\ cmol} = CEC\ cmol_c/kg$$

(2)

Sample concentration of $Sr^{2+}$ was converted to CEC from sample weight, and the atomic weight and valence of $Sr^{2+}$. Each sample treatment was performed in triplicate, with blanks measured below the detection limit (0.05 mg/L).

## Statistical analyses and figures

Statistical analyses were performed in R 4.4.3 to evaluate differences in particle size distribution and CEC across polymer types and degradation times. We compared PE and PET treatments separately for significant differences in particle size distribution with degradation time using one-way ANOVA and Tukey HSD post-hoc tests ($p < 0.05$). ANOVA for particle size included 111–861 measurements per treatment for PE and 107–215 for PET, reflecting the variation in particle counts due to fragmentation during degradation. The Shapiro-Wilk test for normal distribution was determined for particle size in the sample treatments which showed improved normality in particle size distribution when log-transformed. The Shapiro–Wilk test indicated a significant deviation from normality ($p < 0.05$), however the $W$ statistic was relatively high ($W > 0.97$), due to the large sample size of particle counts. The PE 2000 hour degraded sample was found to contain NPs (<1 µm) present outside of our detection limit of the microscope; therefore, we fitted a truncated normal distribution to log-transformed particle size data using lower truncation point at log(1) (the detection limit) to the observed particle size distribution and predicted the entire normal distribution with the estimated mean and standard deviation. We also used the estimated parameters to compute the cumulative probability below the detection limit as an estimate of the proportion of NPs. Uncertainty around this estimate was calculated using a non-parametric bootstrap with 1000 resamples.

One-way ANOVA, followed by Tukey's HSD post-hoc tests, was used to compare PE and PET at each pH level and degradation time for significant differences in CEC ($p < 0.05$). The Shapiro-Wilk test confirmed a normal distribution for CEC samples of PE and PET at each pH level and degradation time ($W > 0.78$, $p > 0.05$). Homogeneity of variances was assessed using Levene's test ("car" package), which showed no significant differences in variances across treatment groups ($F = 1.67$, $p > 0.05$).

## Results

### Particle size distribution of degradation MPs

To understand how UV degradation affects MPs particle size, we measured the size of particles for PE and PET: 1) exposed for 400 hours in the degradation chamber, 2) exposed for another 2000 hours, and 3) a set of non-degraded, pristine MPs (0 hours) (UV exposure in de-ionized water, stirred). The resolution of the transmitted light microscope was 0.56 µm per pixel, therefore a lower limit of particle detection was set at 1 µm.

For PE particles, a large decrease in particle size with degradation time was observed (Fig 1). ANOVA of PE treatments showed a significant reduction in particle size over degradation time ($F = 2531$, $p < 0.001$; log-transformed to normalize distribution). Tukey HSD post-hoc tests indicated no significant difference in particle size between pristine PE (0 hours) and degraded PE at 400 hours, but a large significant reduction at 2000 hours ($p = 1.9 \times 10^{-13}$; see S2 Fig). The mean particle size for PE was 375 µm, 370 µm, and 8 µm at degradation times of 0, 400, and 2000 hours, respectively (see S1 Table). Combining histograms and Gaussian graphs showed that with increased degradation, the frequency of larger particles decreased while the frequency of smaller particles increased. The decrease in particle size was accompanied by a narrowing of the particle size distribution width (Fig 1). The starting observed particle range for PE was 161–689 µm. After 2000 hours of degradation, PE then ranged from below the detection limit (< 1 µm) to the largest detected particle at 66 µm. Therefore, to assess the amount of NPs produced which we could not detect under the microscope, we fitted a truncated normal distribution, which estimated 2.5% of particles (<1 µm) are missing from our observed size measurements (see S3 Fig), with a 95% confidence interval of 1.7–3.5% obtained via non-parametric bootstrap. The sharp reduction in particle size of PE suggests accelerated fragmentation with degradation time, and the truncated normal distribution estimation statistically indicates a likely transition into the nanoplastic size range with further degradation.

For PET particles, degradation led to a significant decrease in particle size at each degradation step from 0 to 400 hours (ANOVA: $F = 25$, $p < 0.001$). Tukey's HSD showed a significant reduction from 0 to 400 hours ($p = 0.01$) and 2000 hours ($p = 1.4 \times 10^{-10}$), given the large number of particle measurements, although only a subtle decrease in particle size

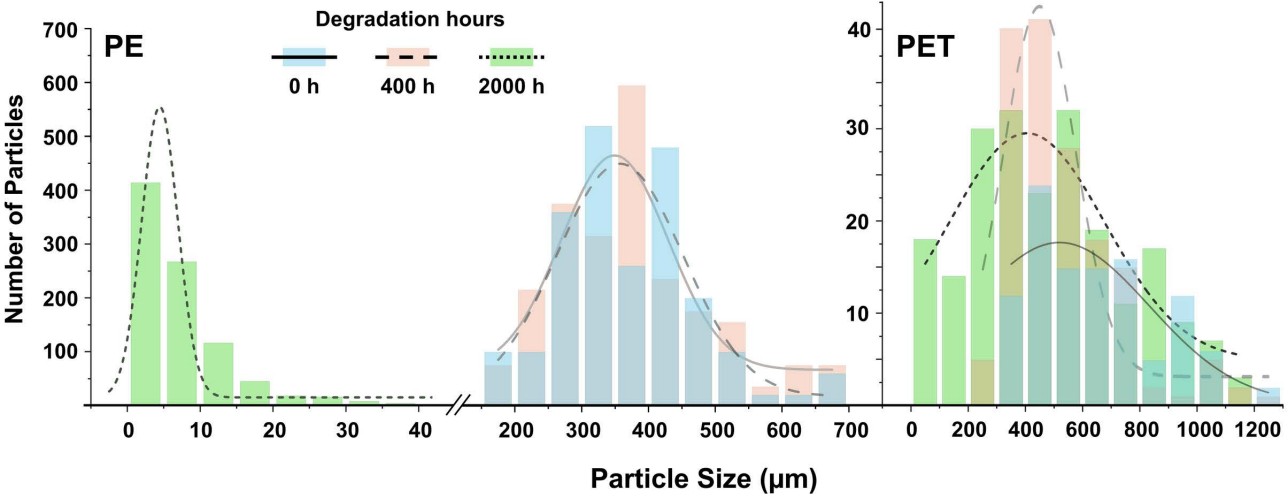

**Fig 1. Particle size distribution of PE (left) and PET (right) microplastics degraded at 0, 400, and 2000 hours (solid and blue, dashed and red, and dotted line and green, respectively); the left y-axis corresponds to PE 2000 h and the right y-axis corresponds to all other treatments.**

was observed, with means of 653 µm, 531 µm, and 484 µm, respectively. However, PET reduced in size to 74% on average at 2000 hours of degradation compared to the large reduction to 2% of the initial size of PE (see S1 Table, S2 Fig). For PET 0 h, particles were in a range of 333–1239 µm, and after 2000 hours degradation, the range was 24–1165 µm.

## Surface characteristics of degraded MPs

Scanning Electron Microscopy (SEM) provided detailed images of the surface structure of MPs, showing features like cracks, holes, and grooves that formed during degradation. For PE particles, SEM images showed a consistent decrease in particle size (Fig 2). During the first 400 hours of degradation, the particles became more rounded and showed cracks and wrinkles in the surface texture compared to pristine PE. Although surface roughness was not quantified, these qualitative observations from SEM images illustrate progressive surface degradation. Fragmentation occurred mainly from 400 to 2000 hours of degradation, with particle size decreasing severely and surfaces turning rougher. After 2000 hours, a significant fragmentation into NPs formed.

For PET particles, the decrease in the size of particles during degradation was not recognizable by visualization alone (Fig 2). Only by measuring the average diameter of particles under a light microscope, the tiny flaky fragments that were detached from the outer layer of larger particles could be recognized. The surface of particles did not change during the first 400 hours of degradation. However, on the surface of PET particles after 2000 hours of degradation, traces of abrasion appeared. The roughness of the surface increased, and consequently, we observed more tiny flaky fragments on the surface, which had the potential of detaching from their larger initial particle. In summary, the size of PET during degradation did not change much; only tiny flaky fragments likely detached from the outer layer of the larger particles (see SEM images, S4 Fig).

ESEM shows wetting as a function of water pressure in the SEM chamber. This allows for qualitative evaluation of hydrophilicity of surfaces in the degraded polymers at 2°C. The smaller the contact angle the more hydrophilic the surface is. An additional kinetic indication for increased hydrophilicity is that less pressure is needed for the observable condensation to start.

For PE, pristine particles without degradation (PE 0) were primarily hydrophobic (Fig 3), with large contact angles (CA) as expected of hydrophobic surfaces, and wetting started at 750–800 Pa. At 400 hours of degradation, PE remained

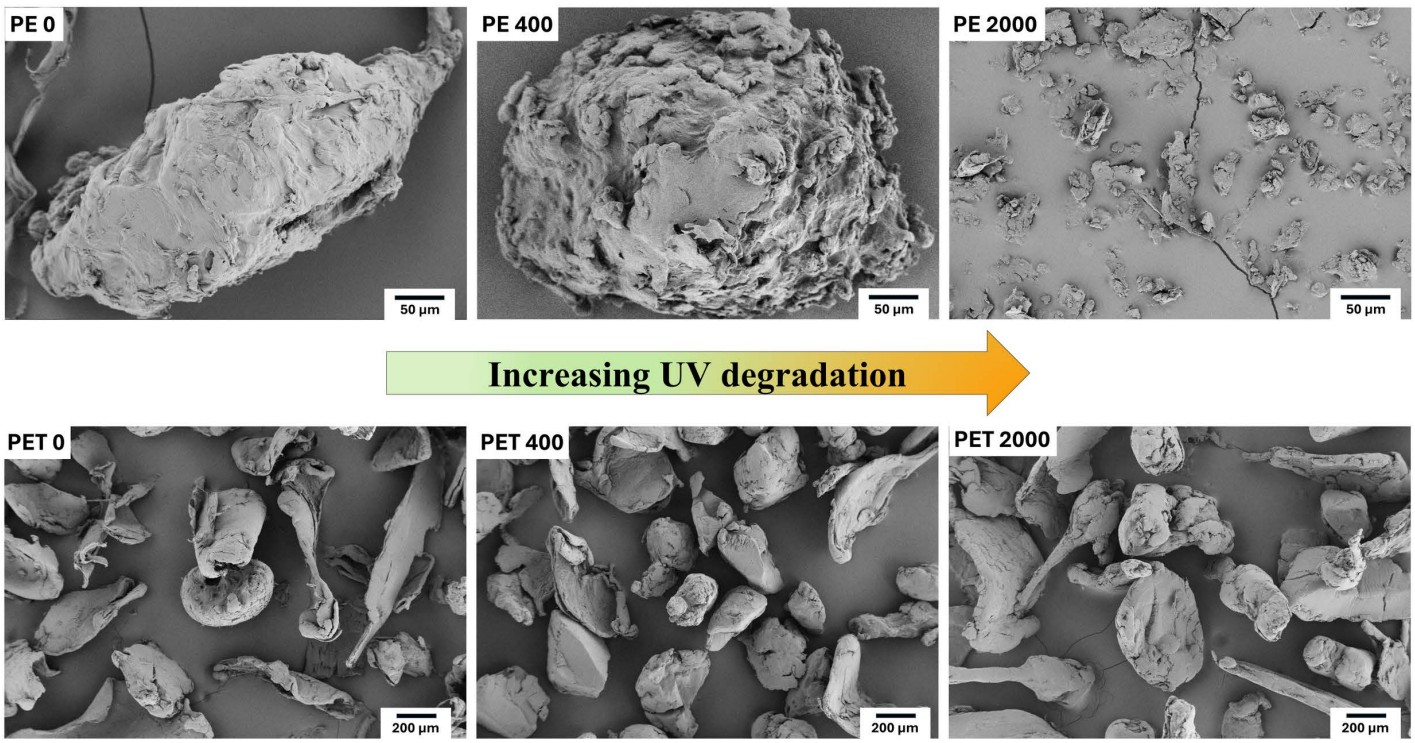

**Fig 2. SEM images of PE (top row) and PET (bottom row) degraded at 0, 400, and 2000 hours.** Degradation decreased the size of PE particles, while PET particles showed minimal size reduction.

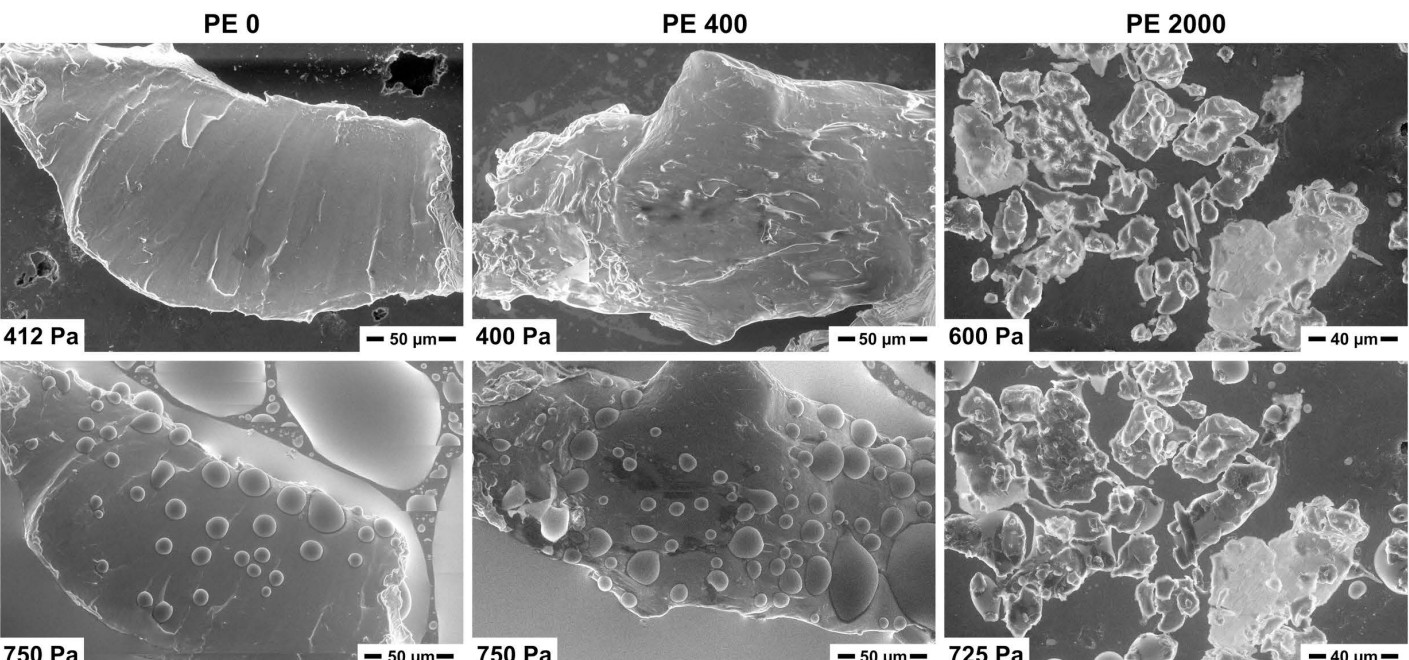

**Fig 3. Representative ESEM images at 2°C of PE at 0, 400, and 2000 hours of degradation.** Especially hydrophobic particles (water droplets with high contact angles) started to drift because of earlier wetting of the carbon support. Degradation changed the surface wetting behavior of PE from hydrophobic to hydrophilic.

mainly hydrophobic, with some areas showing small CA, e.g., particle gaps filling; wetting starting earlier at 730 Pa indicating less hydrophobicity than pristine PE. At 2000 hours, PE became hydrophilic, with low contact angles, gaps filling, and particles surrounding wetting regions; wetting started immediately at 720 Pa and before the graphite support became wetted.

For PET, all kinds of contact angles were present: hydrophilic and hydrophobic areas. There was no significant difference between pristine and degraded particles (S5 Fig).

## Changes in surface chemistry of degraded MPs

The FTIR spectra of pristine PE at 0 hours and degraded PE at 400 hours were similar (Fig 4), with only a small peak appearing for PE at 400 hours at wavenumber 1714 cm$^{-1}$, indicative of carbonyl (C=O) group formation. The increased broadening in the carbonyl peak of PE at 2000 hours up to 1750 cm$^{-1}$ indicates further ester and ketone formation. In PE at 2000 hours, not only did the peak at wavenumber 1714 cm$^{-1}$ increase, but we also observed an increase in transmittance and the appearance of new peaks from 850 cm$^{-1}$ to 1300 cm$^{-1}$, which could indicate a variety of carbon-oxygen single bonds (C-O) as carboxylic acids, ethers, alcohols, and peroxides. The FTIR spectra for pristine and degraded PET showed no differences (Fig 4), although the peak heights for PET at 400 hours were lower than the others.

XPS analysis of the surface polymer chemistry (≈5 nm) of pristine and degraded PE and PET was performed. XPS quantitated and compared atomic ratios of elements to carbon which appeared on the plastic surface (Table 1). After degradation, the surface of PE increased in atomic concentration of oxygen (O), silicon (Si), and fluorine (F), with a subsequent reduction in carbon (C). Pristine PE revealed some initial O concentration bound to the polymer surface, which more than doubled after 400 hours, and further increased with 2000 hours of degradation (Table 1). Si was detected on pristine and degraded PE as C-Si organic bonds (Fig 5), its presence likely due to Si-containing lubricants used in the polymerization process, which subsequently increased with degradation. PE in its structure consists only of C and H, but

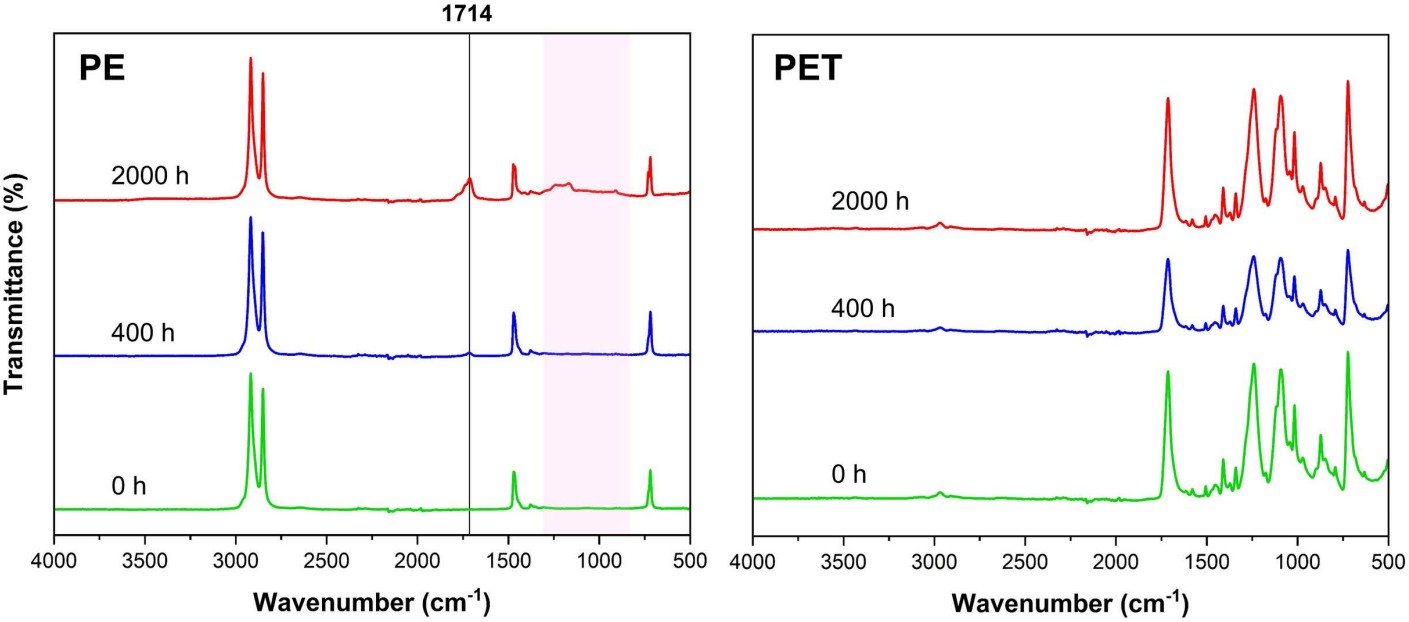

**Fig 4. FTIR spectra of PE (left) and PET (right) at each degradation time; new peak marked at wavenumber 1714 cm$^{-1}$ in PE after 2000 hours of degradation (top).**

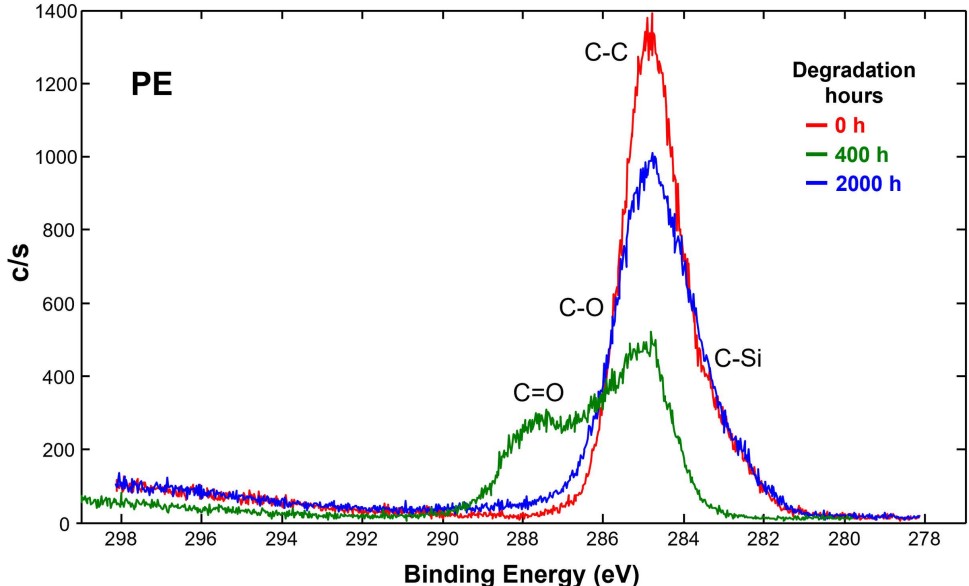

**Fig 5. XPS spectra overlay of carbon C1s spectra, showing change in functional group formation of PE during degradation.**

impurities are often present even in specialized production. Traces of F were also detected after degradation, which was not present in pristine PE, and are attributed to surface contamination from minor abrasion of PTFE-coated magnetic stir bars used for prolonged stirring during degradation.

In XPS, the different binding states of the carbon can be analyzed by the deconvolution of the corresponding high-resolution element spectra (curve fitting) of the C1s spectra (Fig 5). On the surface of PE, the C1s spectra consists mainly of C-C bonds (eV = 284.5) with a tailing to C-Si bonds (eV = 283) (Briggs & Beamson, 1992). At 400 hours of degradation, PE exhibits a strong shift away from C-C bonds on the surface and towards C-O (eV 286) and C=O carbonyl (eV 287–288) and carboxyl (eV = 289) bonds. At 2000 hours, the PE surface recovers its C-C bonds closer to when it was pristine, however with wider tails exhibiting C-O formation. This suggests functional group formation on plastic polymer surfaces is dynamic with degradation time, as size fractionation of particles reveals fresher surfaces.

PET contains C and O, single and double bonds in its polymer structure. After degradation, no increase in O concentration was found in PET (Table 1); instead the appearance of organic C-F bonds (F1s, eV = 689, not shown), similar to those in PE, is attributed to trace PFTE contamination, as chemical fluorination is not expected under the experimental conditions. However, contrary to PE, initial Si content decreased over degradation time on the surface of PET and was not detected after 2000 hours. The C1s spectra of the pristine PET surface shows C-O and C=O and other carboxyl and ester groups (S6 Fig). After degradation at 400 hours, a more pronounced shift to C=O bonds and reduction in C-O bonds occurs, but after 2000 hours of degradation these peaks recover. While slight dynamic changes to surface chemical functionality occur with PET over degradation time, these changes are minimal compared to PE.

## Zeta potential and isoelectric point

Surprisingly, all IEP values were lower than what we would normally expect for non-polar surfaces (pH 4, S1 Fig), with the IEP of pristine PE and PET around pH 3, possibly as the increase in surface area and roughness from the milling process to produce MPs.

The IEP of non-degraded PE (0 h) was around pH 3, with the plateau having high absolute zeta potential values, and after artificial degradation, the IEP decreased (Fig 6). This change seemed to reach its lowest point after long degradation times. In the pH range below pH 4.5, close to the IEP, the zeta potential values were less than ±30 mV. In non-degraded PE, as pH increased, the absolute zeta potential values steadily increased. However, around neutral pH, the rate of increase slowed down, and zeta potential values rose smoothly. In PE 400 h, the decreasing trend changed to an increasing trend at around pH 7.5 as the pH increased. PE 2000 h followed a similar trend at around pH 5.5; and as the pH increased beyond 6, we observed an increase in the zeta potential. We attribute this behavior of the zeta potential to UV-induced swelling of the PE polymer. We assume that UV degradation of PE creates polar groups in the polymer, enabling water to swell the polymer at high pH. This swelling reduces the charge density on the PE surface and thus the absolute value of the zeta potential.

The IEP of non-degraded PET was similar to PE, also with high absolute plateau values and a decreased IEP after artificial degradation. In the pH range below pH 5, close to the IEP, the zeta potential values were less than ±30 mV. In

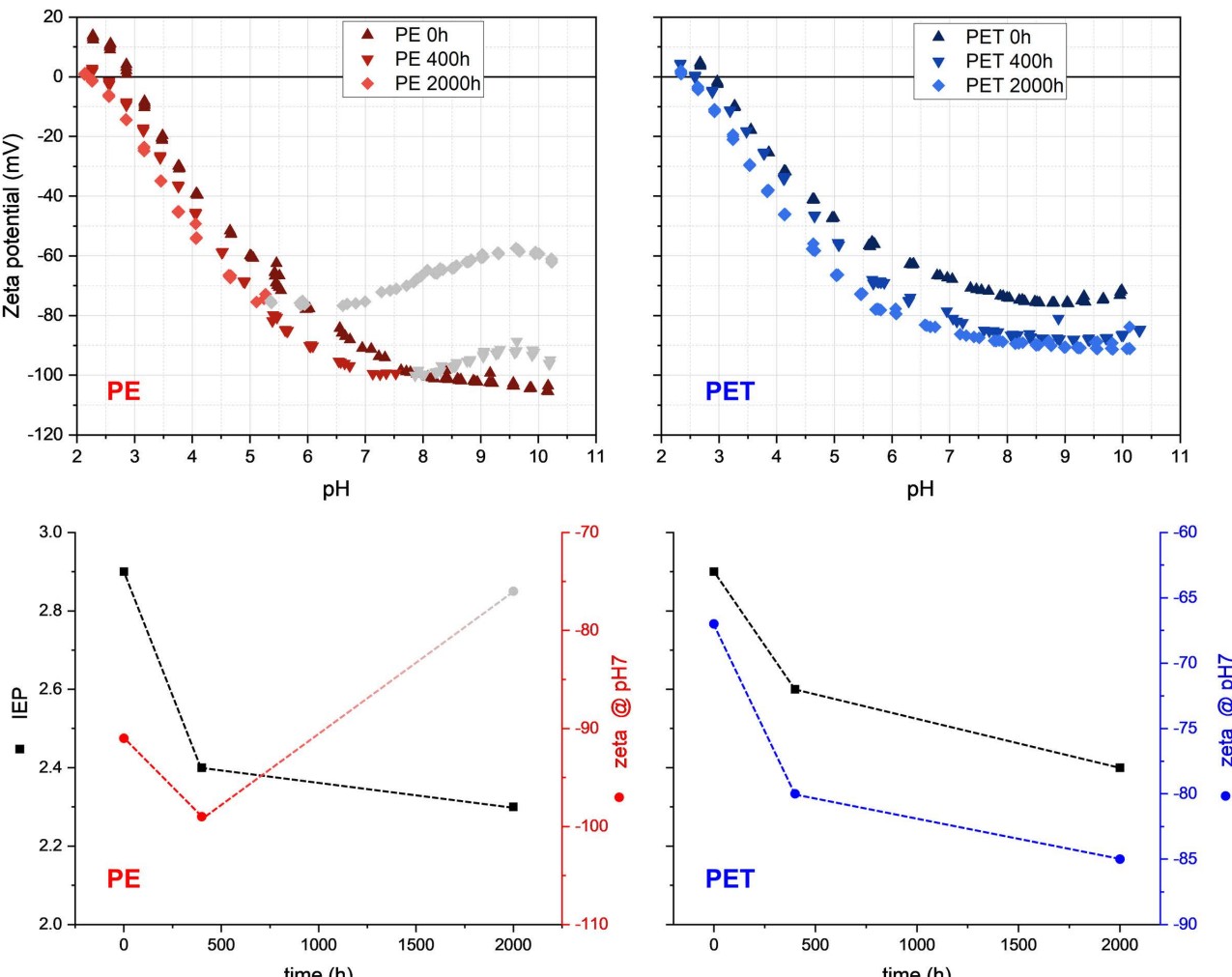

**Fig 6. Effect of degradation state on zeta potential values (top) and isoelectric point (IEP, bottom) of PE (left) and PET (right) at different degradation times.** Gray symbols in PE degraded plastics showed instability in the alkaline region.

non-degraded PET, as pH increased, the absolute zeta potential values steadily increased. However, above neutral pH, the rate of increase slowed down. Both PET 400 h and PET 2000 h showed similar trends to non-degraded PET but with higher absolute zeta potential values.

**Cation exchange capacity of degraded MPs particles**

Degradation of PE led to a significant increase in CEC values dependent on pH (ANOVA: $F = 325$, $p < 0.001$). The CEC values for both non-degraded PE (PE 0) and PE degraded for 400 hours (PE 400) were very low and close to the control sand, meaning they were near zero charge from pH 4–7 (Fig 7). At pH 9, the CEC values for both PE 0 and PE 400 increased slightly to about $0.8 \pm 0.2$ $cmol_c$/ kg (mean$\pm$s.d.), significantly above the control at pH 9 (Tukey's HSD: $p < 0.005$). However, for PE exposed for 2000 hours (PE 2000), significant differences in CEC occurred compared to control at pH 7 ($p = 5.7 \times 10^{-5}$) and pH 9 ($p = 1.1 \times 10^{-12}$). The charges on PE 2000 varied and increased with an increase in pH; at low pH, PE 2000 had a net negative CEC because it had more positive charges than negative ones. As the pH increased to 7, the CEC values rose to about $0.9 \pm 0.1$ $cmol_c$/ kg in the positive range of PE 2000. When the pH was increased to 9, the CEC dramatically rose to mean value of $7.5 \pm 0.5$ $cmol_c$/ kg. To better understand the absolute CEC values of degraded MPs, we compared to the CEC of montmorillonite, one of the most reactive soil constituents. Using montmorillonite as a reference, at pH 9, the CEC of PE degraded for 2000 hours was about one-tenth of montmorillonite's adsorption capacity ($77.3 \pm 7.7$ $cmol_c$/ kg). Even at pH 7, the CEC of 2000 h PE was about one-seventieth of montmorillonite's adsorption capacity.

Although to a lesser degree than PE, degradation of PET also led to a significant increase in CEC values dependent on pH (ANOVA: $F = 26$, $p < 0.001$). From pH 4–7, the CEC values for non-degraded PET (PET 0), PET 400, and PET 2000

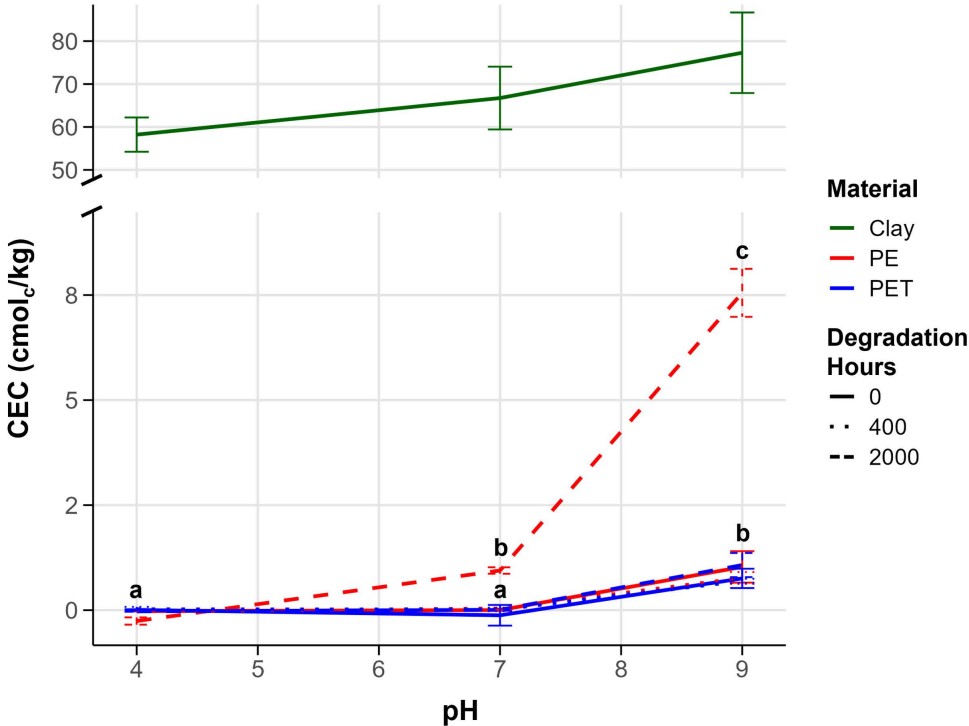

**Fig 7. CEC values for pristine and degraded PE and PET, and montmorillonite clay at three pH levels with standard deviation and significance letters from Tukey HSD post-hoc tests.**

were very low and close to the control sand, meaning near zero ($p = 1$). When the pH was increased to 9, the CEC values for both non-degraded and PET 400 significantly differed from control ($p < 0.05$) and reached a mean of $0.7 \pm 0.1$ cmol$_c$/ kg. However, for PET 2000 h, the CEC slightly increased to $1.1 \pm 0.2$ cmol$_c$/ kg compared to control ($p = 5.4 \times 10^{-6}$). Comparing the absolute CEC values of PET with montmorillonite showed that PET had very low CEC values, around one-seventieth of montmorillonite's adsorption capacity, even after 2000 hours of degradation (Fig 7). This demonstrates that longer degradation times increase surface reactivity, especially in alkaline environments, but the extent of reactivity is dependent on plastic polymer type.

## Discussion

### Reduction of PE and PET particle size and increase in surface area due to UV degradation

UV degradation significantly impacts the particle size distribution of PE, whereas for PET this is less clear. It was shown before that the degree of crystallinity plays a crucial role in forming microcracks on the surface of MPs [15,16], with lower crystallinity leading to more microcracks appearing earlier during degradation. Changes in MPs particles during degradation varied greatly depending on the composition and degree of crystallinity of the polymers. Menzel et al. [16] concluded that the break-up of PE is driven by surface fragmentation. Indeed, we could show that PE particles significantly decreased in size after 2000 h of UV degradation, an average 46x smaller in magnitude on average compared to starting particles. For PE, fragments of various sizes were seen on the particle surfaces after 2000 degradation hours. The particles were very small but had sharp edges, and secondary NPs were attached to larger particles (Fig 2). A significant number of PE NPs particles formed after 2000 hours of degradation, an estimated 2.5% (see S3 Fig), which would predict a larger pool of NPs formed from sunlight exposure also in the environment with respective consequences for their accumulation and reactivity in soil [24]. To date, field quantification of NPs in soils is lacking due to analytical limitations [25]. However, several laboratory studies demonstrate polymer fragmentation to the nanoscale under environmental weathering conditions [8,16,26,27], while field studies have shown widespread microplastic accumulation in soils [28,29]. This suggests that a nanoplastic yield of a few percent is environmentally plausible in terrestrial systems.

Degradation and fragmentation should also affect the roughness of MPs. Accordingly, Menzel et al. [16] observed that the first 400 hours of PE degradation resulted in smoother surfaces of PE due to abrasion, but did not reduce particle size. However, in our study, we did not observe significant smoothing of PE during the first 400 hours; instead, the particles became more rounded (Fig 2). Fragmentation was the main process from 400 to 2000 hours, with particle size decreasing exponentially and surfaces turning rougher, indicating rapid break-up, which was consistent to Menzel et al.'s observations. In our study, we did not consider degradation of PE after 2000 hours because, according to Menzel et al. [16], after 2000 hours particle sizes remained stable. Particle disintegration continued mainly at the nanoscale, but a plateau occurred as disintegration was balanced by agglomeration (NPs particles sticking to larger ones). The substantial reduction in size of degraded PE and increased surface area may exhibit increased reactivity in soils, and the formation of NPs have largely unknown consequences but may be more environmentally relevant due to their highly changed properties.

For PET, fragmentation is less likely to occur, as photodegradation and the associated decrease in mechanical properties have been shown to be limited to a thin affected layer, about 15 μm, and chemical changes were detectable only within the top 50 μm surface of PET [30–32]. The limited surface penetration of UV radiation and oxygen into the bulk of the material explains why deeper layers remained relatively unaffected. The degradation products formed at the surface act as a barrier, preventing further degradation effects in the bulk [33]. In our study, PET particles showed some reduction in size at each degradation step, however with only tiny flaky fragments detaching from the surface. For PET particles, the decrease in the size of particles during degradation was not recognizable in SEM images (Fig 2); only by measuring the average diameter of particles under a microscope, the tiny flaky fragments that were detached from the outer layer of larger particles could be recognized (S2 Fig).

UV exposure of PET leads to significant changes in surface morphology, such as increased roughness and the formation of microcracks, which were evident in the SEM images. These changes were accompanied by variations in color and gloss, which are more noticeable in the early stages of degradation [34]. However, we did not observe significant changes in color or the formation of microcracks until 2000 hours of accelerated degradation, where traces of abrasion first appeared. The roughness of the PET surface also increased (S4 Fig) and consequently, we observed more tiny flaky fragments on the surface, which had the potential of detaching from their larger particle. These are aligned with previous research on PET degradation, indicating that photodegradation primarily affects the surface of the polymer [31,32]. The lack of significant particle size reduction in the initial 400 hours suggested that surface abrasion did not substantially reduce size early on. However, after 2000 hours, the increased roughness and tiny flaky fragments indicated more advanced degradation that could continue with increased degradation time. In the soil environment, the fragmentation of PET is very slow compared to PE, leading to prolonged accumulation and persistence in soils [35]. Over time, the increased surface roughness of degraded PET may interact with soil mineral and microbial components, affecting long-term soil dynamics.

## Surface chemical changes in degraded PE and PET

Degradation of MPs in the environment, particularly under sunlight exposure, initiates change in their surface chemistry. UV irradiation, a key component of sunlight, induces alterations in the polymer structure. For PE, this process introduces carboxyl, hydroxyl, hydroperoxides and oxygen containing functional groups, ultimately leading to chain scission [6,7,36,37]. Consequently, as a result of this chemical degradation process, the physicochemical properties of PE are transformed, resulting in an increase in the presence of functional groups on PE polymers [8,38].

Indeed, as seen here in the FTIR and XPS spectrum of degraded PE, especially for PE exposed for 2000 hours, new functional groups such as carboxylic acids were observed. Specifically, C-O stretching vibrations, characteristic of oxidation products such as alcohols, carbonyl compounds, esters, ethers, or ketals/acetals. The presence of these peaks suggested the formation of new functional groups due to the degradation process, indicating chemical changes on the surface of PE which will likely change its reactivity in the environment [16]. It is likely not possible to incorporate much more than 10% oxygen into the surface, as the oxidation is not controlled during degradation: in a first step, hydroxyl groups (alcohols) are formed, a portion of them are oxidized to ketones, then forming carboxylic acid groups, and eventually volatilizing as $CO_2$ [39,40]. This is perhaps why our PE degraded for 400 hours showed a high increase in surface oxidized functional groups in the XPS (Fig 5), before major fractionation at 2000 hours into smaller sizes revealed new surfaces, therefore functional group formation over degradation time is dynamic with particle size reduction and surface exposure [16]. Degradation of PE at 2000 hours caused more fragmentation into smaller pieces, creating new submicron surfaces with higher surface areas, which increased its reactivity in combination with functional group formation (see summary schematic, S7 Fig). The increased surface area, decreased hydrophobicity, and introduction of hydroxyl, hydroperoxides, and other polar functional groups can affect pH-dependent charges, such as surface chemistry in soil minerals and organic matter.

All measured samples for both PE and PET showed strongly negative zeta potential values because of large hydrophobic surface area without water adsorption (Fig 6). As the size of PE particles decreased, the absolute zeta potential values increased. This happened because smaller particles have a larger surface area, leading to higher zeta potential values [6,26,41]. Generally, when we increased the pH, the absolute zeta potential values of MPs also increased. The specific reaction of degraded PE particles to alkalinity could be interpreted as a swelling of the surface (gray symbols, Fig 6), however, this would need further investigation. ESEM imaging showed a shift from hydrophobic behavior in pristine PE to hydrophilic after degradation (Fig 3). This indicated a shift in polarity and the ability of PE to interact with the polar phase of water, which is confirmed in recent studies also showing hydrophilic changes to PE artificially or naturally degraded [27,42]. Over time, the increased interaction of PE with water would change its chemical behavior in soils, suggesting changes in plastic chemical sorption over degradation time based on the change from hydrophobic to hydrophilic

interactions. PE that has undergone significant degradation could then interact with pollutants, nutrients, and influence soil pH by interaction with H+ and OH- species in water.

For PET, other studies have shown that UV degradation was more subtle and limited to the surface [32,33,43]. PET contains ester and aromatic groups that could become more oxidized to introduce carbonyl or hydrolyze ester bonds to form carboxyl and hydroxyl groups. This happens because acidic groups might form or settle on the particle surface [6,36,37]. The height of the initial peaks in the spectra of pristine PET at 1711 cm$^{-1}$ (carbonyl stretching) and 1233 cm$^{-1}$ (C-O stretching) increased with increasing degrees of degradation [44]. In our study, PET did not exhibit a large change in chemical functional groups after degradation. While FTIR did not detect functional group change in the top surface (few micrometers), XPS revealed an initial shift away at the surface (few nanometers) after 400 degradation hours, from oxidized carbon to more single carbon bonds (see S6 Fig). At 2000 hours the shift remained the same, suggesting early transformation after 400 hours of the very top surface of PET which becomes stabilized. XPS analysis revealed only slight changes in atomic ratios between C and O (see Table 1). Therefore, we could conclude that the stable chemical properties of PET were resistant to photooxidation even after accelerated degradation, indicating slow transformation and degradation in natural environments. Hydrophobic properties of PET did not change due to degradation but rather showed variable hydrophilic and hydrophobic on the surface of PET. As the chemical degradation of PET had only little effect on its surface chemistry, then the main factor increasing the CEC values was the increasing surface area (Fig 7). The stability of PET may also be attributed to the presence of stabilizers or other additives, which can inhibit photooxidation by absorbing UV radiation [44]. As our PET originated from clean recycled bottles, it likely contained additives typically used in manufacturing that may have contributed to the limited surface transformation observed here. Altogether, size and surface charge analyses indicated that as PE and PET become break down and become smaller in the environment, their reactivity increases, and this reactivity should be most pronounced in alkaline environments.

## Potential roles of degraded MPs particles in soils

Research has shown that PE can affect soil physicochemical properties, with PE significantly increasing CEC [45,46]. However, our findings highlighted that the degradation state of PE played a more crucial role in influencing CEC than merely the presence of PE. Generally, the CEC of most soils increases with pH. At very low pH values, the CEC is usually low. In these conditions, only the permanent charges of clay minerals and a small portion of the pH-dependent charges on colloids hold exchangeable ions. If the input of PE with pH-dependent, variable charge to a soil system is high, especially in limed soil or alkaline soils, this could result in significant changes to nutrient and pollutant cycling. Some research even suggests that increasing MPs in soil could positively affect their fertility by increasing CEC values [47]. However, to give a theoretical calculation example: the measured 77 cmol$_c$/ kg clay for a highly reactive clay such as montmorillonite that might occur to 10% or more in soil would give a total charge of 7.7 cmol$_c$/ kg soil. A soil contaminated with plastic, even under critical management conditions such as ploughing and direct incorporation of PE, might reach 1% plastic content [28,48]. With a measured charge of around 7.5 cmol$_c$/ kg for degraded PE in alkaline conditions, this would contribute 0.075 cmol$_c$/ kg of soil, a small but potentially relevant contribution. If continued input and accumulation of plastic pollution to soil continues, clay content is reduced or absent in soil [49], or hotspots occur in specific soil regions where MPs might accumulate >1%, the surface reactivity of degraded PE could start affecting cationic nutrient and pollutant dynamics to a degree comparable or in competition to clay minerals.

Research should focus now on micro-sites in soil where plastic surface charge can matter either due to a massive enrichment compared to bulk soil [29], or due to the attraction of soil microbes that are part of carbon and nutrient cycling. The degraded and charged MPs may then play a role for micro-scale processes such as soil aggregation and, indirectly, for nutrient dynamics. Additionally, it is important to focus research on assessing the ratio between fresh and degraded portions of PE and PET and further plastic types in soil, as this might affect the interpretation of hydrophobic and reactive chemical effects which would be especially relevant for soils under climatic stress or drought conditions.

## Conclusion

This study provided information on the physicochemical changes of PE and PET MPs during UV degradation. Studying the degradation of MPs is crucial because degradation alters their physicochemical properties, including size, surface area, charge, and chemical composition, influencing environmental reactivity. PE particles showed a significant decrease in size over 46x after 2000 hours of degradation and formed a substantial amount (estimated 2.5%) of NPs < 1 μm after 2000 hours of degradation, which has largely unknown environmental consequences. The formation of new functional groups in degraded PE, such as carbonyl groups, carboxylic acids, and its shift to hydrophilic behavior indicate that degraded PE is more environmentally relevant than pristine PE, as these many changes in surface properties have potential to complex with other soil components, mineral and organic, which require further study to elucidate. In contrast, PET particles showed minimal changes in size, surface chemistry, and CEC, highlighting differences in degradation behavior based on polymer structure and crystallinity. Degradation increased the negative surface charges on both PE and PET, but degraded PE showed much higher reactivity under alkaline conditions, exhibiting a major shift in chemical behavior due to its changed surface with increased hydrophilicity and CEC, indicating greater adsorption capacity which is relevant to alkaline soils or soils with low reactive mineral contents. The degradation rate of PE plays a more crucial role in influencing CEC than merely the presence of PE. Although degraded MPs remain less reactive than natural soil components such as clay, their accumulation and persistence in agricultural soils highlight the need for further investigation of their long-term interactions and effects on soil chemical functions and fertility.

## Supporting information

**S1 Table. Particle size distribution table of PE and PET after 0, 400, and 2000 hours of degradation.**
(PDF)

**S1 Fig. Zeta potential (ζ) of particles with different functional groups in the presence of KCl under different pH conditions.** For non-polar/ non-dissociating surfaces, the isoelectric point (IEP) is determined around pH 4.
(PDF)

**S2 Fig. Particle size distribution of PE and PET plastics after degradation (log-transformed).** Significant differences from Tukey HSD tests are displayed as different characters.
(PDF)

**S3 Fig. PE 2000 hours degraded observed particle size distribution (log-transformed; black line), with estimated fit of truncated normal distribution (red dashed line) which estimated 2.5% of the distribution as undetected particles <1 μm.**
(PDF)

**S4 Fig. SEM images of PE (top two rows) and PET (bottom two rows), pristine (0 h) and degraded at 400 h and 2000 h (left to right), at 5000x and 1000x magnification.**
(PDF)

**S5 Fig. ESEM images of PET pristine (left) and degraded at 2000 hours (right), at 300x magnification.** PET particles show no significant changes in wettability over degradation time.
(PDF)

**S6 Fig. XPS overlay of C1s spectra of PET degraded at 0 (red), 400 (green), and 2000 (blue) hours.**
(PDF)

**S7 Fig. Summary of PE degradation after 2000 hours of accelerated UV exposure (approx. 14 months natural sunlight) with proposed functional oxidation of polymer groups shown below.** Functional group formation

progresses from alcohols to carbonyl and carboxylic acids, with decreased particle size and hydrophobicity, and increased surface area, roughness, charge, and ion binding capacity.
(PDF)

## Acknowledgments

We thank these members of the University of Bayreuth: N. Meides (Macromolecular Chemistry) for providing degraded microplastics, M. Obst of the Analytical Chemistry Lab (CAN) for chemical analysis, A. Jakobs for assistance with microscopic analysis (Ecological Microbiology), M. Löder for conducting FTIR analysis (Animal Ecology I), and M. Becevic (Soil Ecology) for graphic design support. Additional thanks to F. Simon of Polymer Interfaces, Leibniz-Institut für Polymerforschung Dresden e.V., for XPS support; and M. Heider for her support with SEM analysis at the Keylab Electron and Optical Microscopy (BPI), Bayreuth Institute of Macromolecular Research (BIMF). A special thanks to T. Glaser for preliminary CEC research and K. Söllner for her technical expertise.

## Author contributions

**Conceptualization:** Ryan Bartnick, Eva Lehndorff.

**Data curation:** Ryan Bartnick, Shahin Shahriari, Günter K. Auernhammer, Ulrich Mansfeld, Lisa Hülsmann.

**Formal analysis:** Ryan Bartnick, Shahin Shahriari, Günter K. Auernhammer, Werner Reichstein.

**Funding acquisition:** Eva Lehndorff.

**Investigation:** Ryan Bartnick, Shahin Shahriari, Günter K. Auernhammer, Ulrich Mansfeld, Werner Reichstein.

**Methodology:** Shahin Shahriari, Ulrich Mansfeld, Werner Reichstein.

**Project administration:** Eva Lehndorff.

**Resources:** Eva Lehndorff.

**Software:** Lisa Hülsmann.

**Supervision:** Eva Lehndorff.

**Validation:** Lisa Hülsmann.

**Visualization:** Ryan Bartnick, Ulrich Mansfeld.

**Writing – original draft:** Ryan Bartnick, Shahin Shahriari.

**Writing – review & editing:** Günter K. Auernhammer, Ulrich Mansfeld, Lisa Hülsmann, Eva Lehndorff.

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
