## [Decision Letter · Decision Letter 0]

17 Sep 2025

Dear Dr. Bartnick,

Thank you for submitting your manuscript to PLOS ONE. After careful consideration, we feel that it has merit but does not fully meet PLOS ONE’s publication criteria as it currently stands. Therefore, we invite you to submit a revised version of the manuscript that addresses the points raised during the review process.

We look forward to receiving your revised manuscript.

Kind regards,

Muammar Qadafi

Academic Editor

PLOS ONE

Journal Requirements:

“This study was funded by the Deutsche Forschungsgemeinschaft (DFG, German Research Foundation) – SFB 1357 – 391977956.”

“This study was performed within the Collaborative Research Center 1357 “Microplastics - Understanding the mechanisms and processes of biological effects, transport and formation: From model to complex systems as a basis for new solutions.” This study was funded by the Deutsche Forschungsgemeinschaft (DFG, German Research Foundation) – SFB 1357 – 391977956. We thank these members of the University of Bayreuth: N. Meides (Macromolecular Chemistry) for providing degraded microplastics, M. Obst of the Analytical Chemistry Lab (CAN) for chemical analysis, A. Jakobs for assistance with microscopic analysis (Ecological Microbiology), M. Löder for conducting FTIR analysis (Animal Ecology I), and M. Becevic (Soil Ecology) for graphic design support. Additional thanks to F. Simon of Polymer Interfaces, Leibniz-Institut für Polymerforschung Dresden e.V., for XPS support; and M. Heider for her support with SEM analysis at the Keylab Electron and Optical Microscopy (BPI), Bayreuth Institute of Macromolecular Research (BIMF). A special thanks to T. Glaser for preliminary CEC research and K. Söllner for her technical expertise.“

“This study was funded by the Deutsche Forschungsgemeinschaft (DFG, German Research Foundation) – SFB 1357 – 391977956.”

Reviewers' comments:

Reviewer's Responses to Questions

**Comments to the Author**

1. Is the manuscript technically sound, and do the data support the conclusions?

Reviewer #1: Yes

Reviewer #2: Yes

2. Has the statistical analysis been performed appropriately and rigorously?

Reviewer #1: Yes

Reviewer #2: Yes

3. Have the authors made all data underlying the findings in their manuscript fully available?

Reviewer #1: Yes

Reviewer #2: Yes

4. Is the manuscript presented in an intelligible fashion and written in standard English?

Reviewer #1: Yes

Reviewer #2: Yes

Reviewer #1: The manuscript by Bartnick et al. investigates how accelerated UV weathering alters the physicochemical properties of microplastics (PE and PET) and assesses implications for soil reactivity.

COMMENTS:

Abstract:

The abbreviation “MP” appears without prior definition. Change to “microplastic (MP)” at first mention.

Keywords:

Add “microplastics” explicitly and ensure consistency (e.g., “UV weathering” versus “ultra-violet”

Introduction :

General comment: The rationale for choosing only PE and PET could be strengthened by citing environmental abundance data in addition with production/use data.

L# 40: The sentence “Plastics effect on soil functions as a foreign reactive surface is not fully understood” is grammatically awkward. Revise to “The impact of plastics as foreign reactive surfaces on soil functions remains poorly understood.”

Results:

L#265 : PET’s minimal size reduction is noted, but the statistical significance (p = 1.4 x 10⁻¹⁰) seems overstated given the small physical change. Clarify the practical relevance.

L#275: SEM images are described well, but the interpretation of “crumpled paper” morphology is subjective. Include quantitative roughness metrics if available.

L#316: XPS data shows meaningful trends, but the presence of fluorine (F) is unexpected. Was contamination ruled out?

L#356: Zeta potential trends are well-documented, but the “strange instability” in PE at alkaline pH needs further explanation or hypothesis.

L# 513 : PET’s surface chemistry is described as stable, yet the introduction of fluorine at 2000 h contradicts this. Clarify whether this is an artifact or a true transformation.

Discussion:

L#418: The estimation of 2.5% NPs formation is intriguing but speculative. Suggest validating with nanoparticle tracking analysis (NTA) or dynamic light scattering (DLS).

In Discussion, expand on environmental relevance of 2.5% NP yield; compare to field studies.

Minor Comments

In FTIR spectra, specify instrument resolution (e.g., 4 cm⁻¹) and number of scans.

Remove redundancy in concluding sentences; the last three sentences largely restate earlier points.

All figures should include clear scale bars and legends.

Finally, the explanation of dynamic functional group formation is excellent. Consider adding a schematic to visualize the degradation pathway. The discussion of PET’s resistance to degradation is well-supported. However, the role of additives or stabilizers in PET should be acknowledged.

Reviewer #2: Summary and general comments

This work studied the effect of weathering on microplastics on their cation sorption capacity and compared it to that of montmorillonite clay. The manuscript is well-written and easy to understand. The results and discussion are clear, and support the conclusion. However, there are a few minor issues. Please see the specific comments below for details.

Specific Comments

1. Recheck journal requirements if the header requires numbering.

2. Abstract. Clarity request for sentence at line 30, where the authors mentioned MP exhibits cation sorption of clay, without mentioning the sorption capacity of clay. In this sentence, only PE is mentioned; however, it is not clear if PET have the same adsorption level.

3. Line 114. Consider adding information such as purity and molecular weight range.

4. Lines 130-131. Is there any reason to choose 400 and 2000 hours of weathering? What do these chosen times represent? Is there any information regarding the equivalent of 1 hr of accelerated weathering equivalent to how much time at the nature?

5. Lines 155. Is this FTIR using an ATR module?

6. Line 221-240. How many data points were involved in the ANOVA?

**Do you want your identity to be public for this peer review?** For information about this choice, including consent withdrawal, please see our Privacy Policy

Reviewer #1: **Yes: ** Gopal Chandra Ghosh

Reviewer #2: No

---

## [Author Response · Author response to Decision Letter 1]

30 Oct 2025

Comments to the Author

Reviewer #1: The manuscript by Bartnick et al. investigates how accelerated UV weathering alters the physicochemical properties of microplastics (PE and PET) and assesses implications for soil reactivity.

COMMENTS:

Abstract:

The abbreviation “MP” appears without prior definition. Change to “microplastic (MP)” at first mention.

Thank you, it has been amended.

Keywords:

Add “microplastics” explicitly and ensure consistency (e.g., “UV weathering” versus “ultra-violet”

Thank you for the recommendation. Keywords have been updated, and “UV degradation” is now consistent throughout the MS.

Introduction :

General comment: The rationale for choosing only PE and PET could be strengthened by citing environmental abundance data in addition with production/use data.

We have included environmental abundance estimates as part of this introductory paragraph:

L#49: “Polyethylene (PE) is the most widely used plastic in the world, accounting for around 30% of all plastics [3]. PE degrades relatively easily when exposed to ultraviolet (UV) light [4]. Polyethylene terephthalate (PET) is also a widespread plastic and of environmental concern due to its strong resistance to degradation [5]. Globally, 79% of produced plastics are estimated to remain on land, accumulating in landfills or the natural environment [3]. When exposed to UV light, PET can break down into smaller particles, but this process is slow, and the particles can persist in the environment. For these reasons, PE and PET were chosen in this study to understand the physicochemical transformations during UV degradation.”

L# 40: The sentence “Plastics effect on soil functions as a foreign reactive surface is not fully understood” is grammatically awkward. Revise to “The impact of plastics as foreign reactive surfaces on soil functions remains poorly understood.”

Thank you for the recommendation, the sentence reads much better.

L#42: “The impact of plastics as foreign reactive surfaces on soil functions remains poorly understood, as the change of plastic surface properties during environmental exposure must be considered.”

Results:

L#265 : PET’s minimal size reduction is noted, but the statistical significance (p = 1.4 x 10⁻¹⁰) seems overstated given the small physical change. Clarify the practical relevance.

Thank you, the low p-value is from the large particle count data set measured. We added the particle counts into the statistical methods section:

L#231: “ANOVA for particle size included 111-861 measurements per treatment for PE and 107-215 for PET, reflecting the variation in particle counts due to fragmentation during degradation.”

We clarified this point further in the results of manuscript:

L#283: “Tukey’s HSD showed a significant reduction from 0 to 400 hours (p = 0.01) and 2000 hours (p = 1.4 x 10-10), given the large number of particle measurements, although only a subtle decrease in particle size was observed, with means of 653 µm, 531 µm, and 484 µm, respectively.”

L#275: SEM images are described well, but the interpretation of “crumpled paper” morphology is subjective. Include quantitative roughness metrics if available.

We appreciate the reviewer’s comment. Quantitative surface roughness metrics were not determined, as this study focused on chemical and compositional changes during polymer degradation rather than surface topography. The description of surface morphology is therefore based on qualitative SEM observations intended to illustrate physical degradation features. We have clarified this point in the revised text to avoid overinterpretation:

L#293: “During the first 400 hours of degradation, the particles became more rounded and showed cracks and wrinkles in the surface texture compared to pristine PE. Although surface roughness was not quantified, these qualitative observations from SEM images illustrate progressive surface degradation.”

L#316: XPS data shows meaningful trends, but the presence of fluorine (F) is unexpected. Was contamination ruled out?

We thank the reviewer for noting the fluorine observation. The F1s peak detected only in 2000 h samples and is consistent with C–F bonding, which is now identified as PTFE contamination in our degradation setup. The microplastic samples were continuously stirred with PTFE-coated magnetic stir bars during degradation; thus, minor PTFE abrasion is the most likely source of surface fluorine contamination. Covalent fluorination under the applied UV/wet conditions is chemically implausible, and XPS instrument blanks and unexposed controls showed no F signal. Because the fluorine signal is confined to the outer surface and appears only after prolonged exposure, it is treated as surface contamination that does not affect the reported oxidation (O/C) trends.

We now make clear in the Methods and Results where the contamination is likely from:

Methods:

L#127: “MPs were maintained at a constant temperature of 38°C, immersed in deionized water, and under mechanical stress from stirring with a PTFE-coated magnetic stir bar.”

L#171: “XPS survey spectra were inspected for adventitious elements, and a low-intensity F1s signal was observed on the subset of 2000 h samples (see Table 1, Results). Given the use of PTFE-coated stir bars during long exposures, PTFE abrasion from stirring is the most probable source of surface fluorine contamination.”

Results:

L#354: “Traces of F were also detected after degradation, which was not present in pristine PE, and are attributed to surface contamination from minor abrasion of PTFE-coated magnetic stir bars used for prolonged stirring during degradation.”

L#371: “After degradation, no increase in O concentration was found in PET (Table 1); instead the appearance of organic C-F bonds (F1s, eV = 689, not shown), similar to those in PE, is attributed to trace PFTE contamination, as chemical fluorination is not expected under the experimental conditions.”

L#356: Zeta potential trends are well-documented, but the “strange instability” in PE at alkaline pH needs further explanation or hypothesis.

Thank you for the comment. We have modified the formulation to avoid any misunderstandings. Our hypothesis is that UV irradiation creates charged groups in PE. At a high enough density, these charged groups cause the UV-degraded PE to swell in water at a high pH. According to this hypothesis, the increase in zeta potential is a consequence of the polymer swelling. As the polymer swells, the density of charged groups at the surface decreases, thereby reducing the absolute value of the zeta potential. This hypothesis is consistent with the observed changes in contact angle (Fig 3) due to UV degradation. The discussion has been amended:

L#391: “In PE 400 h, the decreasing trend changed to an increasing trend at around pH 7.5 as the pH increased. PE 2000 h followed a similar trend at around pH 5.5; and as the pH increased beyond 6, we observed an increase in the zeta potential. We attribute this behavior of the zeta potential to UV-induced swelling of the PE polymer. We assume that UV degradation of PE creates polar groups in the polymer, enabling water to swell the polymer at high pH. This swelling reduces the charge density on the PE surface and thus the absolute value of the zeta potential.”

L# 513 : PET’s surface chemistry is described as stable, yet the introduction of fluorine at 2000 h contradicts this. Clarify whether this is an artifact or a true transformation.

As stated before, this is an artifact from the PTFE-coated magnetic stir bars and not chemical transformation. This has been amended in the Discussion:

L#555: “XPS analysis revealed only slight changes in atomic ratios between C and O (see Table 1). Therefore, we could conclude that the stable chemical properties of PET were resistant to photooxidation even after accelerated degradation, indicating slow transformation and degradation in natural environments.”

Discussion:

L#418: The estimation of 2.5% NPs formation is intriguing but speculative. Suggest validating with nanoparticle tracking analysis (NTA) or dynamic light scattering (DLS).

In Discussion, expand on environmental relevance of 2.5% NP yield; compare to field studies.

Thank you for the recommendation. DLS measurements were taken early on but proved to be inconsistent below the micrometer range for our samples. We attempted DLS measurements again with an alternative method (volume and count, and using surfactant and ultrasonic), but the NPs remained undetectable possibly due to agglomeration and the large range of particle sizes (DLS measurement data will be uploaded to Zenodo for reference). We do not have an NTA available to us. Therefore, we chose an estimation based on the expected Gaussian distribution of particle sizes under controlled formation conditions, which we see with all other degraded plastics we observed (Fig 1) and expect this distribution to extend to a lower range. To further clarify, we additionally provide now a confidence interval (95%) estimation for the 2.5% NPs yield, which resulted in a 1.7-3.5% range (1000 resamples via non-parametric bootstrap).

Methods:

L#242: “We also used the estimated parameters to compute the cumulative probability below the detection limit as an estimate of the proportion of NPs. Uncertainty around this estimate was calculated using a non-parametric bootstrap with 1000 resamples.”

Results:

L#271: “Therefore, to assess the amount of NPs produced which we could not detect under the microscope, we fitted a truncated normal distribution, which estimated 2.5% of particles (<1 µm) are missing from our observed size measurements (see S3 Fig), with a 95% confidence interval of 1.7–3.5% obtained via non-parametric bootstrap.”

In the discussion, we expanded on the relevance to the environment. Given the limited information and analytical constraints on quantification of nanoplastics in soils, previous laboratory studies and field observations of microplastics suggest nanoplastics in terrestrial environments are very plausible.

Discussion:

L#454: “To date, field quantification of NPs in soils is lacking due to analytical limitations [25]. However, several laboratory studies demonstrate polymer fragmentation to the nanoscale under environmental weathering conditions [8,16,26,27], while field studies have shown widespread microplastic accumulation in soils [28,29]. This suggests that a nanoplastic yield of a few percent is environmentally plausible in terrestrial systems.”

Minor Comments

In FTIR spectra, specify instrument resolution (e.g., 4 cm⁻¹) and number of scans.

Thank you for the comment, the resolution was 4 cm-1 with 8 sample scans and 8 background scans. The methods for FTIR now include these details:

L#161: “The FTIR measurements covered a range of wavenumbers from approx. 4000 to 400 cm⁻1 with a resolution of 4 cm⁻1, averaging 8 sample scans and 8 backgrounds scans.”

Remove redundancy in concluding sentences; the last three sentences largely restate earlier points.

Thank you for the recommendation, the conclusion has now been shortened and removed redundant points:

L#600: “This study provided information on the physicochemical changes of PE and PET MPs during UV degradation. Studying the degradation of MPs is crucial because degradation alters their physicochemical properties, including size, surface area, charge, and chemical composition, influencing environmental reactivity. PE particles showed a significant decrease in size over 46x after 2000 hours of degradation and formed a substantial amount (estimated 2.5%) of NPs <1 µm after 2000 hours of degradation, which has largely unknown environmental consequences. The formation of new functional groups in degraded PE, such as carbonyl groups, carboxylic acids, and its shift to hydrophilic behavior indicate that degraded PE is more environmentally relevant than pristine PE, as these many changes in surface properties have potential to complex with other soil components, mineral and organic, which require further study to elucidate. In contrast, PET particles showed minimal changes in size, surface chemistry, and CEC, highlighting differences in degradation behavior based on polymer structure and crystallinity. Degradation increased the negative surface charges on both PE and PET, but degraded PE showed much higher reactivity under alkaline conditions, exhibiting a major shift in chemical behavior due to its changed surface with increased hydrophilicity and CEC, indicating greater adsorption capacity which is relevant to alkaline soils or soils with low reactive mineral contents. The degradation rate of PE plays a more crucial role in influencing CEC than merely the presence of PE. Although degraded MPs remain less reactive than natural soil components such as clay, their accumulation and persistence in agricultural soils highlight the need for further investigation of their long-term interactions and effects on soil chemical functions and fertility.”

All figures should include clear scale bars and legends.

All figures have been updated with clear scale bars and legends and in the appropriate format.

Finally, the explanation of dynamic functional group formation is excellent. Consider adding a schematic to visualize the degradation pathway. The discussion of PET’s resistance to degradation is well-supported. However, the role of additives or stabilizers in PET should be acknowledged.

Thank you for the suggestion, a schematic has been added to the supporting information to compliment the discussion. Additionally, a discussion of PET stabilizers has been added:

Discussion:

L#522: “Degradation of PE at 2000 hours caused more fragmentation into smaller pieces, creating new submicron surfaces with higher surface areas, which increased its reactivity in combination with functional group formation (see summary schematic, S7 Fig).”

L#562: “The stability of PET may also be attributed to the presence of stabilizers or other additives, which can inhibit photooxidation by absorbing UV radiation [44]. As our PET originated from clean recycled bottles, it likely contained additives typically used in manufacturing that may have contributed to the limited surface transformation observed here.”

Supporting Information:

“S7 Fig. Summary of PE degradation after 2000 hours of accelerated UV exposure (approx. 14 months natural sunlight) with proposed functional oxidation of polymer groups shown below. Functional group formation progresses from alcohols to carbonyl and carboxylic acids, with decreased particle size and hydrophobicity, and increased surface area, roughness, charge, and ion binding capacity.”

Reviewer #2: Summary and general comments

This work studied the effect of weathering on microplastics on their cation sorption capacity and compared it to that of montmorillonite clay. The manuscript is well-written and easy to understand. The results and discussion are clear, and support the conclusion. However, there are a few minor issues. Please see the specific comments below for details.

Specific Comments

1. Recheck journal requirements if the header requires numbering.

Thank you, after reformatting to the PLOS One manuscript requirements, it seems to suggest line numbers starting from the title header.

2. Abstract. Clarity request for sentence at line 30, where the authors mentioned MP exhibits cation sorption of clay, without mentioning the sorption capacity of clay. In this sentence, only PE is mentioned; however, it is not clear if PET have the same adsorption level.

We appreciate the reviewer’s comment and have revised the sentence in the Abstract for clarity, including CEC for clay, PE and PET:

L#33: “Especially for degraded PE incorporated in soil, the alteration of its surface can exhibit comparatively one-tenth the cation sorption power of clay in alkaline environments (≈7.5 vs. 77 cmolc / kg at pH 9), while degraded PET remained comparatively low (≈1.1 cmolc / kg).”

3. Line 114. Consider adding information such as purity and molecular weight range.

Thank you, the technical data sheets of the plastics have been uploaded to the Zenodo database and referenced in the methods:

L#117: “Further technical data sheets of plastic polymers are available online (see Data Availability).”

4. Lines 130-131. Is there any reason to

---

## [Decision Letter · Decision Letter 1]

5 Nov 2025

UV-degraded polyethylene exhibits variable charge and enhanced cation adsorption

PONE-D-25-42187R1

Dear Dr. Bartnick,

We’re pleased to inform you that your manuscript has been judged scientifically suitable for publication and will be formally accepted for publication once it meets all outstanding technical requirements.

Kind regards,

Muammar Qadafi

Academic Editor

PLOS ONE

Additional Editor Comments (optional):

Reviewers' comments:

Reviewer's Responses to Questions

**Comments to the Author**

Reviewer #1: All comments have been addressed

2. Is the manuscript technically sound, and do the data support the conclusions?

Reviewer #1: Yes

3. Has the statistical analysis been performed appropriately and rigorously?

Reviewer #1: Yes

4. Have the authors made all data underlying the findings in their manuscript fully available?

Reviewer #1: Yes

5. Is the manuscript presented in an intelligible fashion and written in standard English?

Reviewer #1: Yes

Reviewer #1: (No Response)

**Do you want your identity to be public for this peer review?** For information about this choice, including consent withdrawal, please see our Privacy Policy

Reviewer #1: **Yes: ** Gopal Chandra Ghosh

---

## [Editor Report · Acceptance letter]

PONE-D-25-42187R1

PLOS ONE

Dear Dr. Bartnick,

I'm pleased to inform you that your manuscript has been deemed suitable for publication in PLOS ONE. Congratulations! Your manuscript is now being handed over to our production team.

Kind regards,

on behalf of

Dr. Muammar Qadafi

Academic Editor

PLOS ONE